# Behavioural determinants of physiologically-relevant light exposure
Anna M. Biller [1,2] ✉, Priji Balakrishnan [3,4] & Manuel Spitschan[1,2,5]

Light exposure triggers a range of physiological and behavioural responses that can improve and challenge health and well-being. Insights from laboratory studies have recently culminated in standards and guidelines for measuring and assessing healthy light exposure, and recommendations for healthy light levels. Implicit to laboratory paradigms is a simplistic input-output relationship between light and its effects on physiology. This simplified approach ignores that humans actively shape their light exposure *through behaviour*. This article presents a novel framework that conceptualises light exposure as an individual behaviour to meet specific, person-based needs. Key to healthy light exposure is shaping behaviour, beyond shaping technology.

Humans have evolved under predictable 24-h light-dark cycles on Earth due to solar radiation. Light is a critical stimulus that humans depend on to navigate in and interact with their everyday environment. But light also influences us in other ways: we need a strong daily light signal to keep our internal circadian clock aligned with the external world[1–3]. Light acts as a *zeitgeber*—a temporal cue—that synchronises internal (body) time and external (environmental) time, a process called circadian photoentrainment. The internal time is given by circadian rhythms (from Latin *circa dies*, about a day), which are intrinsic 24-h oscillations in physiological and behavioural processes such as sleep-wake cycles, hormone secretion, metabolism and immune responses, and cognitive functions such as attention[1–3]. There is also inter-individual variability in the timing of entrainment leading to a distribution of early to late chronotypes in the population[3–6].

More detailed investigations of the effects of light beyond vision only began in the 1980s. The so-called non-visual effects of light, or non-image-forming (NIF) responses to light, include diverse functions such as alertness, mood, well-being, mental health, sleep, and circadian rhythms[1,7]. In particular, the effects of light on sleep and circadian rhythms have gained significant attention due to the emergence of LED lighting and hand-held light-emitting devices in our modern lifestyles in combination with low-illuminated offices and ubiquitous indoor work. With increased light exposure at the wrong time, particularly during evening or night hours, there is growing concern about the mismatch between internal and external time and subsequent adverse health consequences[1,7]. Misaligned circadian rhythms and sleep disturbances have been associated with an increased risk of several chronic health conditions, including metabolic disorders, cardiovascular diseases, mood disorders, and certain types of cancer[1,7,8]. Too

little light in combination with near work has also been proposed as a likely mechanism underlying the development of myopia, a growing population health concern especially in Asia[9–11].

Laboratory data on the effects of light on circadian physiology and sleep have culminated in recent recommendations for healthy light exposure in everyday life prescribing target light levels[12]. These recommendations, and the underlying data, broadly suggest that humans should experience *bright days* and *dark nights*. Current evidence, however, is limited in its ecological validity, as typical laboratory stimuli contain minimised stimulus features, thereby isolating, e.g., spectrum, wavelength, or intensity. This is in stark contrast to the rich visual world around us varying in space and time. Consequently, it is not clear to what extent laboratory-based recommendations for minimum and maximum light levels apply to real-world exposures. To add to this complexity, individual light exposure can vary significantly with life circumstances such as location, urbanicity, meteorological conditions, type and time of day, time of year, type of work (e.g., shift work, indoor/outdoor work), indoor environments (e.g., space with or without windows), lifestyle factors (e.g., habits, hobbies), and age, gender, or chronotype[13–17]. There is also increasing evidence that light sensitivity varies significantly between individuals[18]. The current recommendations do not factor in such diversity in the response to light.

Additionally, it is an open question how target light levels can be reached from a behavioural point of view. This article presents a novel framework in which light exposure is viewed as an active and individual behaviour. By considering the 'ingredient' behaviour, light exposure behaviour can be studied in a more integrative way (i.e., embedded within the psychology and behaviour of an individual and their daily life and culture) and across disciplines (e.g., psychology, behavioural and vision sciences,

¹Department Health and Sport Sciences, Chronobiology & Health, TUM School of Medicine and Health, Technical University of Munich, Munich, Germany. ²Translational Sensory & Circadian Neuroscience, Max Planck Institute for Biological Cybernetics, Tübingen, Germany. ³Laboratory of Architecture and Intelligent Living (AIL), Karlsruhe Institute of Technology, Karlsruhe, Germany. ⁴Chair of Lighting Technology, Technische Universität Berlin, Berlin, Germany. ⁵TUM Institute for Advanced Study (TUM-IAS), Technical University of Munich, Garching, Germany. ✉e-mail: anna.biller@tum.de

ethology, and chronobiology). Health psychology and behavioural sciences can help with designing interventions that change behaviour into habits with the long-term aim to deliver precision behavioural health and medicine. Only by delivering personalised interventions at the right time for the right person in their right (cultural and situation) context behaviour change can be sustained over a longer timescale, and even constantly adapted within one's lifetime.

## From 'passive' light exposure to human-light interactions

### A brief review of ocular and retinal mechanisms

When light enters the eye and reaches the retina, it interacts with photoreceptor cells by changing their conformation enabling phototransduction, a process through which light energy is converted into neural signals (Fig. 1A). These signals are processed in a series of retinal cells, eventually reaching the optic nerve, which carries them to the brain's visual cortex and other areas for signal processing, integration, and interpretation (Fig. 1B, C). The retina contains two 'canonical' photoreceptor classes, the cones and rods. The cones are responsible for our vision of colour, space, and motion. There are three types of cones (S cones, M cones, L cones), each containing a specific light-sensitive pigment that is most sensitive to either short (peak at ~440 nm), medium (peak at ~530 nm), or long wavelengths of light (peak at ~558 nm) respectively[19]. Cones are densely concentrated in the fovea, the central part of the retina. The rods are more sensitive to light than cones but do not provide colour vision. They are specifically sensitive to low-light or dim conditions and contain a different light-sensitive pigment than the cones. Rods are more abundant in the peripheral regions of the retina[20].

Only about 25 years ago, a third class of photoreceptors, the intrinsically photosensitive retinal ganglion cells (ipRGCs), was discovered in the retina[21–23]. These ipRGCs are especially sensitive to short-wavelength light due to the photopigment melanopsin and mediate the NIF effects of light by transduction via axons to the hypothalamus, which contains the suprachiasmatic nucleus (SCN, Fig. 1C) that orchestrates the peripheral clocks in the rest of the body (Fig. 1D)[22–24]. In addition to their intrinsic photosensitivity, the ipRGCs also receive input from the rods and cones. As a consequence, all photoreceptors could potentially contribute to the non-visual effects of light; it has also been demonstrated that melanopsin signals also reach the primary visual cortex (V1) where they are thought to partially influence visual perception[7,25–28]. Nevertheless, the ipRGCs are most relevant for the NIF effects of light, including melatonin suppression and circadian photoentrainment.

### Non-image-forming (NIF) effects of light

A main consequence of human exposure to the right light (quality, quantity) at the right time (of day, across the year) is circadian photoentrainment, which describes the synchronisation of internal (circadian) time and external (clock) time[29]. Melatonin, a hormone released by the pineal gland in the evening and at night, signals the body to rest and sleep at darkness thereby serving as a circadian signal in this synchronisation process[30]. Melatonin naturally starts to be released several hours before habitual sleep onset, reaches a peak of release in middle of the subjective night and stops being released after sleep offset. Since light suppresses the production of melatonin[31–34], the *timing* (when)[35,36], *duration* (how long)[37,38], *temporal pattern* (how light is structured over time, e.g., flickering)[39–41], *intensity* (how much illuminance)[34,42], and *spectrum* (which wavelengths and spectral composition)[32,33,38,43,44] of light exposure can influence photoentrainment. Depending on the specific combination of these light properties, light exposure can subsequently lead to circadian phase delays and later sleep timing, phase advances and earlier sleep timing, or even no phase shifting at all with no effect on sleep timing[7,42,45,46]. These effects have been formalised in the phase-response curve (PRC) to light, which describes the magnitude and direction of individual phase shifts as a function of the individual light exposure[35,36]. PRCs show that for most people, the phase advance window lies in the 'biological' morning while the phase delay window is in the evening hours.

Light also directly impacts sleepiness and alertness[47–49] through ascending arousal systems in the hypothalamus reaching the cortex (e.g., tuberomammillary nucleus, locus coeruleus, raphe nuclei)[50]: bright high-melanopic light acutely increases alertness levels while dim low-melanopic light contributes to sleepiness thereby serving as an important modulator for cognitive performance. This arousal system is influenced by inhibitory actions of the ventrolateral preoptic nucleus (VLPO): during wakefulness, arousal systems inhibit the VLPO, while during sleep, the VLPO inhibits these arousal systems to maintain sleep states. There is also emerging evidence that light exposure prior to bedtime or even during the day can also *directly* affect sleep quality and sleep architecture (i.e., amount of specific sleep stages)[51]. Other NIF effects of light include pupil constrictions (through the ipRGCs' trajectory to the olivary pretectal nucleus which sends signals to the autonomic nervous system), heart rate and core body temperature modulation (through the autonomous nervous system), and cortisol production (through the SCN's effect on the hypothalamic-pituitary-adrenal axis)[52].

### Measuring the ipRGC-influenced responses to light

Since the combination of these five light exposure parameters (timing, duration, temporal pattern, intensity, and spectrum) play a major role in shaping our physiological output, quantification of these attributes of light is necessary to measure their NIF effect, as in real-world scenarios, light exposure patterns are highly complex[53]. Traditionally, photopic illuminance has been used to describe light intensity in practical lighting design or architecture. Illuminance measured in lux [lx] is a metric for the amount of light reaching a specific surface per unit area and weights the spectrum by the sensitivity of the L and M cones. Given that activities such as reading, working, or navigating were the primary focus of lighting research and application, illuminance used to be a suitable metric for characterising the dependence of visibility, visual acuity, or visual comfort on light intensity.

With the discovery of ipRGCs and their contributions to physiology in the early 2000s and onwards, it became clear that photopic illuminance is not physiologically relevant for the NIF effects[7]. Consequently, the importance of spectral sensitivity, timing, duration, temporal pattern, intensity, and spectrum of light in influencing these NIF effects has now become the focus of attention. Lucas et al.[52] developed α-opic radiance and irradiance as novel metrics which take the special sensitivity profile of ipRGCs into account (i.e., how ipRGCs respond to light across various wavelengths and intensities), quantified by different spectral sensitivity functions. This is done by weighting the light spectrum by the spectral sensitivity of the respective photoreceptors[19]. This approach has been standardised in a SI-compliant system by the International Commission on Illumination (CIE) in 2018[19] in which effective rates of photon capture for each of the human retinal opsins under a given light condition are equated to the photopic properties (e.g., illuminance) of a standard 6500 K(D65) daylight spectrum that would produce the same rate of photon capture (D65 is a standard illuminant that represents the average colour of natural daylight at noon, assuming a clear sky). This approach defines for each photoreceptor class the α-opic equivalent daylight illuminance (EDI; where α-opic denotes one of the five human opsin classes that can contribute to ipRGC-influenced responses)[19]. The melanopic equivalent daylight illuminance (mEDI, measured in lux) allows for a better prediction of the NIF effects of light as it reflects the weighting by the melanopsin spectral sensitivity curve. In brief, melanopic EDI is a more accurate and physiologically motivated measure to capture NIF effects of light for humans.

In addition to these novel light metrics, reporting standards have been established to better compare light studies and their outcomes on physiology. Spitschan and colleagues recently provided an overview of necessary standards to report light exposure in laboratory studies[54]. These include the spectral power distribution of the stimulus from the observers' point of view, the background light environment, and reporting of the α-opic light levels[54]. However, so far mostly illuminance levels, duration of light (exposure), and sometimes light colour (e.g., blue, amber) are reported when describing light interventions. To allow for richer descriptions of light interventions, the

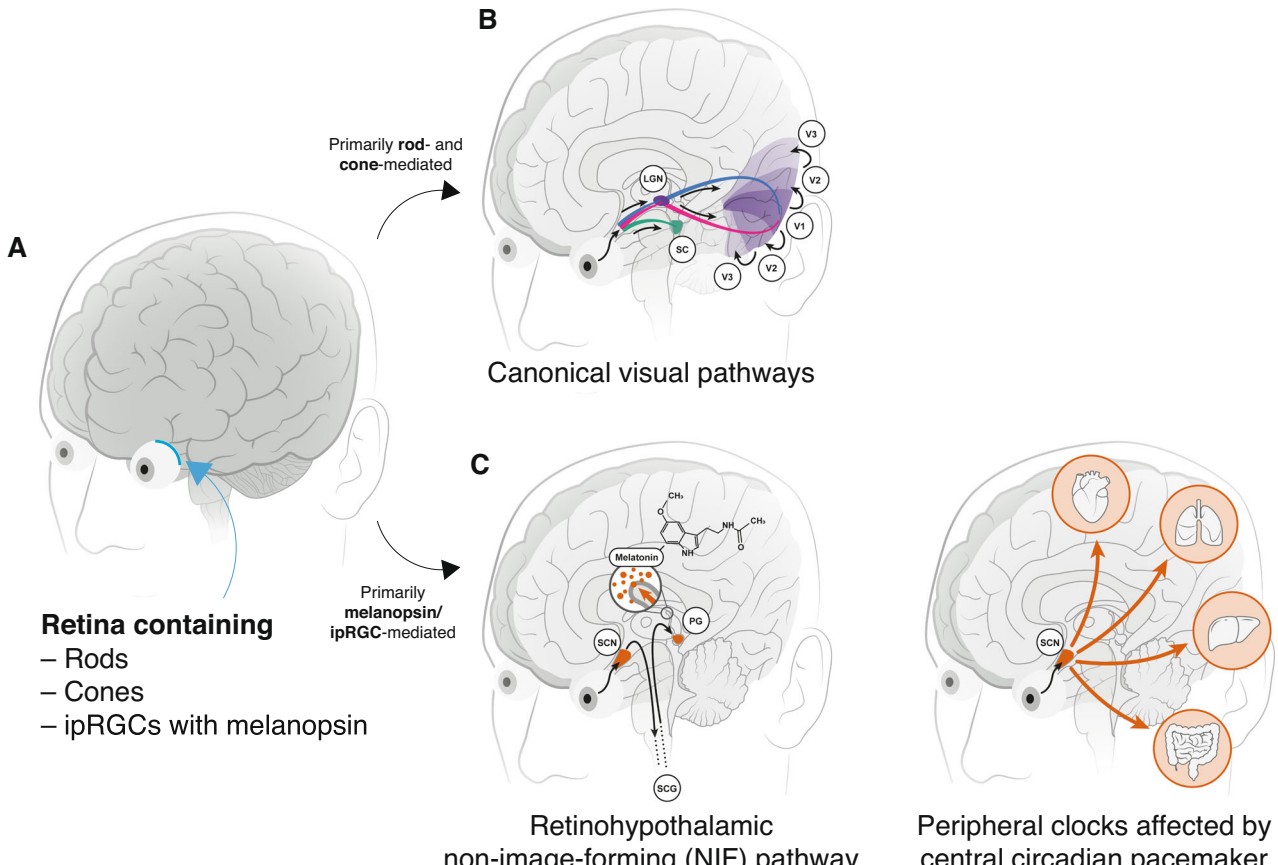

**Fig. 1 | Overview of the visual and non-image forming pathway in humans.**
**A** *Retinal photoreceptors*. Schematic of the brain with the retina at the back of the eye (blue) containing the rods, cones, and the intrinsically photosensitive retinal ganglion cells (ipRGCs) expressing the photopigment melanopsin. **B** *Canonical visual pathways*. Diagram of the *canonical* rod- and cone-mediated visual pathway. Rod and cone signals from the retina relayed to the lateral geniculate nucleus (LGN) from which they further travel to the primary visual cortex (V1), and then higher visual areas (V2, V3). Information from the right visual field in each eye is relayed to the left visual cortex, and vice versa (blue/pink). A second pathway (green) relays light directly to the superior colliculus (SC). **C** *Retinohypothalamic non-image-*

*forming pathway*. The melanopsin/ipRGC-mediated pathway connects the retina to the hypothalamus, and more specifically, the suprachiasmatic nucleus (SCN). The pathway from the SCN to the pineal gland involves a multi-synaptic route: the signal travels from the SCN to the intermediolateral cell column of the spinal cord, then to the superior cervical ganglion (SCG), which is part of the sympathetic nervous system. From the SCG, the signal reaches the pineal gland (PG) to trigger melatonin release in response to darkness. **D** *Peripheral clocks affected by the circadian pacemaker*. Downstream peripheral clocks, e.g. located in the heart, liver, kidney, or colon are also affected by the central circadian pacemaker, the suprachiasmatic nucleus (SCN).

ENLIGHT Checklist was developed in a modified Delphi exercise for standardising the reporting of light interventions[55].

## Novel recommendations for healthy light exposure

Improved quantification of NIF effects has paved the ground for a consensus statement developed by an international team of experts to translate laboratory evidence into light exposure recommendations to best support physiology, sleep, and wakefulness in healthy adults[12]. The recommendations include a minimum melanopic EDI of 250 lux at eye level during daytime, a maximum of 10 lux at least 3 h prior to bedtime (in residential/indoor settings), and a maximum of 10 lux of ambient melanopic EDI at nighttime for the sleep environment (Fig. 2, dashed lines).

While these recommendations represent an excellent evidence-based starting point, the laboratory evidence on which these recommendations are based is limited in ecological validity which is reflected in the ambiguous and impractical recommendations presented in this consensus statement. Firstly, there is no easy way to determine and measure the melanopic EDI in everyday life for the end-user nor easily for the researcher (in contrast to methods used for lab animals such as telemetric monitoring and infrared observations[56]). Secondly, real-life conditions are quantitatively and qualitatively more complicated than laboratory conditions. Typical stimuli used in laboratory studies contain

minimised stimulus features to allow the isolation of light features. In real-life, however, the qualities of light (timing, duration, temporal pattern, intensity, and spectrum) are not isolated from each other and the environmental scenes we perceive are spatially and temporally highly complex and quite unlike conditions used in laboratories.

Typical real-world patterns in a Western 24/7 society are, for example, *dim days* and *bright nights*: this includes the exposure to dim light during the day (e.g., due to electrical lighting in office buildings below the recommend 250 lux mel EDI), and relative bright, blue light after sunset (usually above 10 lux mel EDI) which suppresses melatonin, delays circadian phase, and consequently also delays sleep (yellow profile of individual 2 in Fig. 2B). On the other hand, a strong zeitgeber strength such as natural daylight at high intensities experienced during the day (*bright days*), preferably earlier in the morning during the phase advance window of an individual, can advance the circadian phase and consequently also sleep, and might even be protective to bright light before sleep. This highlights the role of an individual's "photic history"[7,57,58] and "spectral diet"[59] which describe the timing of light as well as the spectral composition of the light an individual experiences across the day respectively (see Fig. 2).

Consequently, it is currently uncertain to what extent laboratory-based recommendations for target light levels are meaningful for the real world or how individuals can easily measure their light environment and determine if

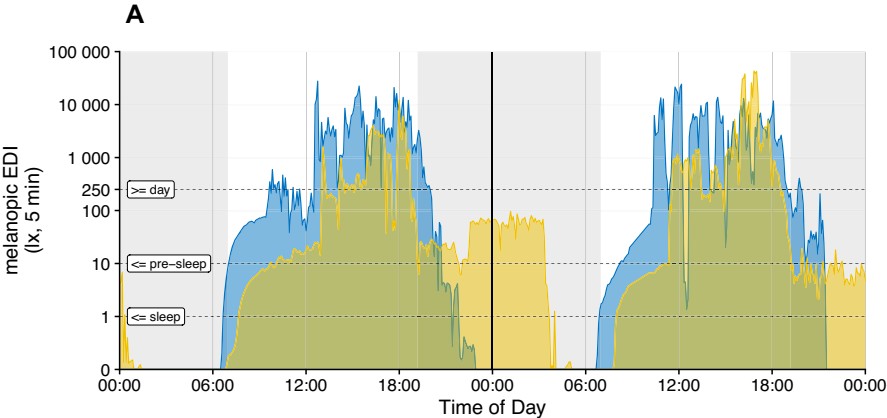

**Fig. 2 | Light exposure profiles from two different individuals. A** Double plot of light exposure profiles averaged over five-minute bins for two individuals (blue vs yellow) over two consecutive days. **B** Quantification for each light exposure profile for the first 24 h. Black vertical line separates day 1 from day 2. Dotted lines show the recommended minimum light levels expressed in melanopic equivalent daylight illumination (melanopic EDI) during sleep (≤1 lx), 3 h before sleep (≤10 lx) and during day time (≥250 lx) according to Brown et al. [12]. The two light exposure profiles in (A) vary drastically and are associated with different behavioural sub-types (yellow vs blue profiles): while both individuals experience the same light onset in the morning around 06:30, individual 1 (blue) receives brighter light earlier in the day (first time above 250 lx mel EDI at 09:40) compared to individual 2 (yellow; at 13:05), spends more time above 250 lx (08:15 vs 04:14), receives brighter light on average (1635 lx vs 334 lx) and experiences the brightest hours earlier in the day (brightest 10 h midpoint at 14:45 vs 17:35). These differences could be due to job situations (e.g., shift work, lighting conditions at work, office vs outdoor job), individual preferences (e.g., sunseeker vs avoider), hobbies (indoor vs outdoor

sports activities) or chronotype (Individual 1 is an intermediate chronotype while individual 2 is classified as late from the Munich Chronotype Questionnaire[6]; MCTQ). In the current example, subjective light exposure data revealed (data not shown) that individual 1 slept until 9:00, spent the morning indoors until 12:00 (daylight indoors), remained outdoors until 21:00 (daylight outdoors), received some electric light indoors until 22:00 and then spent the night sleeping without any reported light source. Individual 1 thus mostly reaches recommended light levels for pre-sleep and during sleep but could receive more bright light early in the morning. Individual 2 (yellow) experienced electric light exposure from midnight to 1:00, spent the night sleeping with light coming in through the window until 14:00, received daylight indoors until 18:00, spent 1 h outdoors in daylight until 19:00 and was inside with electric light until midnight. Individual 2 thus receives light above the recommended levels for pre-sleep and the sleep environment and receives too little light in the morning. Data based on a data collection in Tübingen, Germany, following the protocol described in Guidolin et al. [109] and visualised with the R package *LightLogR*[110].

and how they should adjust their behaviour accordingly. It becomes clear, though, that individually tailored approaches to reach the target levels are necessary for different behavioural subtypes.

**What is light exposure as a behaviour?**

Light exposure and its effects are usually conceptualised as a simple input-output relationship where light enters the eye and produces a specific physiological output through a series of mechanisms (retina → LGN → V1; retina → retinohypothalamic tract → hypothalamus; Fig. 1). We challenge this traditional conceptualisation here and advance the view that humans act within and interact with their light environment to meet specific needs (also see Box 1) that go beyond vision. Pragmatically, we adopt a common definition of behaviour to be "anything a person does in response to internal or external events" defined by "[a]ctions [which] may be overt (motor or verbal) and directly measurable, or covert (activities not viewable, e.g., physiological responses) and indirectly measurable; behaviours are physical events that occur in the body and are controlled by the brain"[60].

In line with this definition, humans do not only perceive light passively but interact with light in various ways: humans design and craft light sources to meet their visual or aesthetical needs or to promote wellbeing and (visual) comfort (e.g., light design, street lighting, dimming, visual illusions, use of candles). We also build environments that shield, integrate, or aesthetically play with light. Furthermore, humans exhibit sun-related behaviours, such as sunlight exposure[61–63] (e.g., sunbathing, tanning, looking towards the sun). There are also light-avoiding reflexes or behaviours (e.g., closing the eyelids, averting eyes away from light source, looking downwards, turning heads[64]) or light control behaviours using objects (e.g., using light switches[65,66], curtains and blinds[67], eye masks, sunglasses, sunscreen use). Humans observe or use light for leisure activities (e.g., sunset watching, stargazing, photography, painting), to navigate around the environment and to increase safety at night. We also use light for work and commercial purposes (e.g., advertisements on screens, illuminated shops at night).

Light also plays a significant part in religion or cultural practices[68], reflected in festivities such as celebration of the summer solstice[69], Saint Lucy's Day in Scandinavia[69], or the festival of Diwali in Hinduism[70,71]. Other practices are weather magic and sun worship[72], use of candles and light in religious ceremonies, or orientation of religious buildings and temples towards the sun[73]. Some cultures also honour solar deities (e.g., Helios in Greek mythology, Amun Ra in ancient Egypt, Apollo in Roman times, Yarhibol in Mesopotamia)[68,69].

On the other hand, the absence of light—darkness—also influence spatio-temporal behaviours, for example, in public spaces like squares or footpaths (e.g., changed perception of safety and reassurance, reduced sta-tionary activities, decreased use of public spaces after sunset)[74–77].

In brief, humans have long been using light to achieve or meet specific needs, have interacted with light sources (electric, sunlight, candles, fires) and crafted and manipulated light. The circadian neuroscience field has yet to appreciate this notion and move beyond a simple input-output relationship.

**A framework for light exposure behaviours**

Figure 3 presents a framework which views human *light behaviour* embedded within the individual's *location* and *culture* and takes place in interaction with the *built environment* (similar to the concept of affordances, see Box 1). While we can also interact and shape the built environment and, at least to some extent also culture, we have little to no impact on external factors such as the location within a time zone (including clock time changes), photoperiod, sunshine hours, local climate, or ambient tem-perature (Fig. 3A). These factors have historically largely influenced and shaped our culture (e.g., customs, festivities, or norms) and have dictated the built environment[78–83] (e.g., building form, windows and openings, proxi-mity to other buildings, street orientations, materiality, or lighting sig-nificance), including its reciprocal interaction[84,85]. Shaped by these determinants are our own behaviours which include how we perceive light

## Box 1 | Human-light interactions as affordances

Human interactions with light can also be thought of as affordances, a concept introduced by psychologist James J. Gibson in the 1970s as part of his ecological approach to perception. Affordances refer to the potential actions or uses that an object (e.g., the light switch), environment, or stimulus (e.g., the light environment, the sun) offers to an individual based on their perception and understanding of its properties to meet certain needs. The concept emphasises the perception-action loop (e.g., the light switch offers the option to turn on, dim or turn off the light at different times for different purposes), highlighting how individuals perceive and understand the world based on their own abilities (e.g., understanding that a switch controls/manipulates light), individual and cultural experiences (attitudes towards light), and context and location (e.g., living outdoors without electricity does not provide the affordances of a light switch; see also Fig. 3)[111,112].

(valence), how we interact with light (affordances), what lifestyle we choose (hobbies, jobs, pets) and what temporal pattern this lifestyle has, or what individual preferences we display towards light (e.g., sunseeker vs sun avoiders). The latter, however, also highlights that within the same culture and location there is room for individual light behaviours as described above enabling unique light interaction behaviours (behavioural subtypes) and (temporal) exposure profiles (Fig. 2). Furthermore, certain disorders or health conditions can interact with individual behaviour and can have a large impact on individual light exposure profiles potentially leading to decreased (e.g., bright light sensitivity, xeroderma pigmentosum, anxiety, depression, severe obesity, gaming addiction, etc.) or increased (e.g., vitamin D deficiency or seasonal affective disorder) sunlight exposure with specific consequences on circadian entrainment.

In the following, two very different examples of location are used to further exemplify this framework. Singapore is an example of a city situated in an equatorial location receiving ample sunlight year-around. However, the region's hot and humid climate largely affects lifestyle, prompting many activities to be conducted indoors in air-conditioned environments[86]. Unlike in northern latitudes such as Europe or North America, sunlight is not considered a scarce commodity at the equator. Instead, due to its abundance and intensity, habitants seek relief from the heat which has also shaped and influenced local architectural practices. In tropical or equatorial locations, traditional houses like *Malay* houses are characterised by sloping roofs with extended overhangs, open-to-sky internal courtyards, a larger number of openings on all sides and an open design. These features efficiently manage heavy rainfall and humidity, increase shade and maximise cross ventilation to cool the interior space[78,87]. Another typical element is a *serambi*, a covered veranda that invites people to sit outside offering shade and cover from rainfall[87] (see also Fig. 2B example of Kerala, India). Given the intense sunlight in these locations, shading elements like coverings or window shutters are integrated to protect buildings from direct sunlight. However, the widespread introduction of air conditioning, the shift to office work and the focus on longer teaching hours in closed rooms to excel in education have led to a decrease in daylight exposure even in equatorial locations (light avoidance behaviour). This decline often falls below recommended levels, despite ample climatic opportunities for exposure year-around, potentially contributing to the rapid rise in myopia[11,88].

The architectural and cultural adaptations to the climate in Scandinavian countries like Sweden and Norway are significantly different from those in tropical and equatorial climates like Singapore[78,79]. The primary challenge in Northern climates is the cold weather, with long winters and sub-zero temperatures. Buildings in these regions are designed to withstand heavy snow loads and provide effective insulation to retain heat[78,89]. To prevent heat loss, buildings are often more compact and sealed which contrasts with the open design and cross ventilation strategies used in tropical climates. However, due to the short photoperiod in winter, there is also an emphasis on maximising natural daylight. Large windows are strategically placed, often facing south, to capture as much sunlight as possible during the day. Scandinavians also embrace winter sports and outdoor activities despite the cold temperatures. Activities such as cross-country skiing, ice skating, and winter hiking provide opportunities for physical activity and to make use of the rare sunlight during the dark winter months (light-seeking behaviour).

Other examples for where locations/geography together with culture and the built environment influences and translates into typical local behaviours are shown in Fig. 3B.

### Towards precision behavioural health

Interventions to promote healthy light behaviour and meet recommended light levels (see also Box 2) must therefore consider the individual and their current lifestyle, personal situation, location, and culture to be maximally effective. However, structural aspects of our living environment, e.g., the architectural or environmental infrastructure, have been the primary targets to achieve certain (health) needs in our context (e.g., changing the built environment to allow for more daylight). Similarly, technology has been developed to better support our physiology, including "human centric lighting"[90], as well as "night shift" technology in light-emitting displays[91]. The behavioural dimension of light exposure remains largely unexplored in interventions. This contrasts with other areas of applications that are unrelated to circadian and sleep health and that aim at *minimising* the harmful effects of sun exposure directly by calling for specific actions or behaviour change (e.g., sunburn protection by efficient use of sunscreen[92] or shielding from sunlight) or by maximising beneficial effects of sunlight (e.g., for myopia prevention[93,94,95] or vitamin D production[96]) or light to increase vision and safety (e.g., street lighting at night influences which behaviours are displayed[75]).

This article presents a novel research and intervention framework that integrates these individual light-interactions and behavioural profiles to harness their unique interventional potential. We build our programme on four pillars (Fig. 4). These are (i) understanding (reasons for) individual light behaviours and exposure profiles, (ii) identifying individual target behaviours and personal barriers, (iii) designing interventions that include behaviour change as a key component in the form of behaviour change techniques, and (iv) deliver these interventions effectively and accessibly including an information-behaviour-feedback loop.

The focus of our programme is to meet light exposure recommendations as formulated by Brown and colleagues[12] (*bright days* and *dark nights*), one target outcome that can be reached by modulating light behaviour in different ways. In general, the programme is also suitable for other behaviour change goals when different behaviours could be addressed (in contrast to e.g., smoking cessation where one specific behaviour needs to be modified).

The first step includes evaluating the individual light interaction behaviour repertoire of an individual (Fig. 4A). This could be achieved through questionnaires such as the Light Exposure Behaviour Assessment (LEBA)[97], or qualitative methods such as interviewing. In theory, such behaviours could also be recorded objectively, for example, through video recording or eye tracking, which is not necessarily feasible and practical for larger-scale interventions. Light exposure profiles as in Fig. 2A can be measured by capturing light logging data but currently face similar practical challenges at scale. Furthermore, the individual trait and state factors modulating sensitivity to light should be described. This includes assessing individual light sensitivity[18,46,98,99] or chronotype[5,15] to understand biological

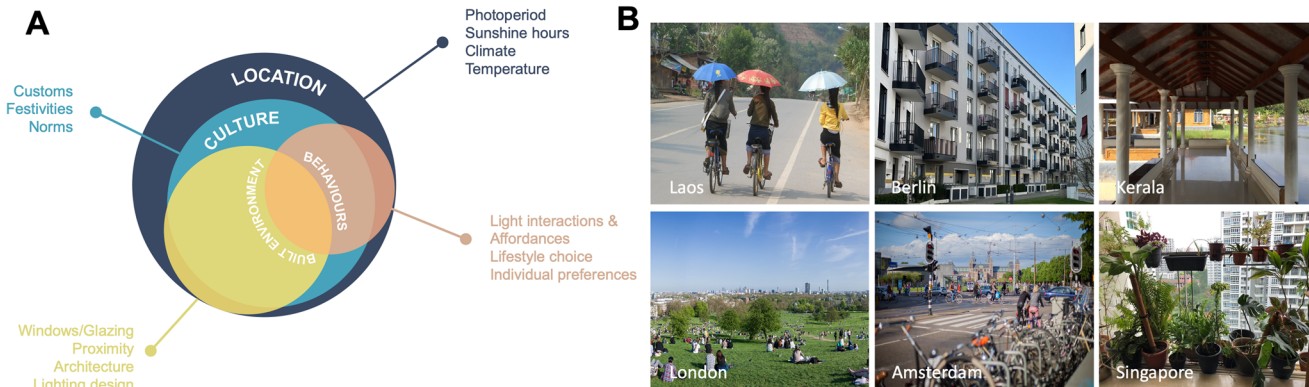

**Fig. 3 | Individual light behaviour is embedded within the location, culture and built environment. A** Individual behaviours related to light (light interactions and affordances, lifestyle choices, preferences) interact with the built environment and are always influenced and shaped by culture and the respective living location. **B** Examples for typical light (interaction) behaviours and their dependence on location include using umbrellas to shield from the sun in the Louang Namthat Province (Laos), open to sky balconies with tables and chairs in in Berlin (Germany), a shaded veranda with seating in Kannur, Kerala (India), meeting for a picnic in

London's Primrose Hill to enjoy the rare sunny weather (UK), commuting by bike in Amsterdam (Netherlands), or caring for plants near a window in Singapore. Photos shown in (**B**) are licenced under CC-BY. Photo from Laos by *recoverling* from https://openverse.org/image/a4c2eba9-915b-49ab-8068-020a492f251e; photos from Berlin, Kerala and Singapore by *Priji Balakrishnan*; photo from London by *Diliff* from https://openverse.org/image/f23f8914-4927-4e18-9df8-ff89c944a1df (CC BY-SA 3.0 licence); photo from Amsterdam by *_dChris* from https://openverse.org/image/67142cba-d655-4c72-851c-9dbffa23be24.

determinants/factors of light behaviour. Since sleep-wake rhythms of late chronotypes are significantly later than their earlier counterparts, they are more likely to be exposed to light at later times of the day as well as more light at night. This might interact with individual light sensitivity such that light at night might be less disturbing for individuals with reduced sensitivity or vice versa. Other characteristics, such as personal circumstances, light preferences, gender roles, age-related life schedules or (ocular) health conditions that cause or are the consequences of specific light behaviours and profiles should also be assessed (Fig. 4A). However, addressing some of these individual phenotypes is currently difficult to assess quickly. While chronotype, or some light behaviours might be estimated subjectively through questionnaires (e.g., MCTQ[6], LEBA[97]), there is currently no easy way to determine light sensitivity for NIF effects of light. Nevertheless, by personalising as much as possible, the intervention is more likely to be effective for the individual and more likely to be sustained over longer periods.

In the second step, a set of target behaviours should be identified together (e.g., reducing evening exposure to bright electric light; yellow profile Fig. 4B) and evaluated with regards to its realistic implementation. Some behaviours might be easier to develop or change than others, hence it might be worthwhile spending some time on identifying these while specifically considering the individual's unique lifestyle and context including their culture and geographical location—and very importantly—personal barriers (Fig. 4B): while certain behaviours are possible to address (e.g., playing video games at night), others are not (e.g., changing the light environment at work or school).

Once this individualisation has taken place, a behaviour change intervention programme can be designed with special focus on the unique needs and behavioural style of the individual (Fig. 4C). The intervention could, for example, include components of cognitive behaviour therapy or more general any kind of behaviour change techniques[60,100], e.g., circadian health education, SMART goal setting (specific, measurable, achievable, relevant, time-based), social support, or changing habits. Some behaviours are one-off behaviours (e.g., setting an automatic night-mode for mobile screen at night) whereas other behaviours could be turned into healthy habits (e.g., working near a window to increase light levels during working hours or always having lunch outside, taking a morning walk etc.) which would need different behaviour change approaches.

Lastly, the intervention needs to be delivered to the individual. Ideally, this should be in an effective, tailored, and accessible way (Fig. 4D). On a simple and large scale, this could be achieved using

chatbots, apps or web-based intervention programmes but of course can also be delivered within a more traditional healthcare setting such as counselling, coaching, or therapy. Smart eHealth[101] interventions, however, have the advantage that they could, in theory, take feedback loops from other information systems, such as upcoming weather information, log light or other behaviour, and integrate this on the go by giving feedback and sending notifications to the user.

## Research questions and future directions towards the behavioural turn

Future studies need to address whether findings from lab studies and their short-term outcomes really translate to long-term health outcomes. Current recommendations are mostly based on lab studies without taking individual light history and diet into account. Humans are not laboratory animals whose light exposure can be controlled carefully for experimental studies[102]. It will thus be difficult to fully characterise our light history and diet in the real-world, especially on larger scales, without developing new light logging devices or other methods which capture this easily, reliably, and less intrusively. While some companies have recently introduced light exposure information measured through smart watches feeding them back to their users (e.g., Apple Watch), this method faces similar barriers as traditional light loggers integrated in actimeters (activity trackers), as they measure light exposure in the wrong place (at the wrist) using irrelevant metrics (lux). In addition, we need to keep improving measurements (metrology) and developing metrics further that quantify NIF effects of light and determine which metrics that describe light exposure are relevant and useful for the individual. Some of these metrics are shown in Fig. 2B but these are only a selection of numerous possible quantifications. This is a field where little standards have been implemented so far which is another challenge to face.

Furthermore, a key priority is to disentangle determinants and moderators of healthy light exposure and their interaction and whether this distinction is relevant for real-life conditions. For example: Is the chronotype-dependent sleep-wake pattern a determinant of an individual light profile, a moderator, or a consequence? Understanding this could help in designing more effective interventions and could be embedded within models or theories of health behaviour[60], such as the health belief model[103] or health action process approach[104].

Since the light interventions as proposed in our programme (Fig. 4) are yet to be evaluated, clinical trials assessing their effectiveness short- and long-term will be necessary. Further studies should also assess if it is possible to combine strategies to address several health endpoints (e.g., improve

## Box 2 | Light interventions to promote wellbeing and health

Given its important role in circadian entrainment and health-related outcomes, light has long been used for therapeutic and well-being purposes. One established area in which light exposure is manipulated is (bright) light therapy (BLT). Historically, BLT has been used for the treatment of seasonal affective disorders (SAD) and its subclinical counterpart (sSAD), depression, bipolar disorder, sleep and circadian-related disorders as well as jet lag prevention or reduction[24,113–117]. Light interventions could theoretically address or manipulate all different parameters of light exposure (timing, duration, temporal pattern, intensity, and spectrum) separately or in combination. Most common manipulations include (blue) light blocking in the evening to avoid melatonin suppression[116,118,119], and/or increasing light exposure either in the morning (i.e., the individual circadian phase-advancing window) to

advance internal time or at times when alertness and vigilance should be promoted (e.g., during shift work)[120–122]. Some studies/interventions also combine these methods or include other interventions, such as melatonin supplementation[123,124], curtailing sleep[125], cognitive-behaviour therapy[126], or manipulating the light environment (dawn-simulating alarms[127–129], use of curtains or blinds[116]). Existing interventions in the circadian field seem to mainly focus on (some of) these four dimensions, without taking the individual light history or spectral diet into account. Recommendations such as those developed Brown et al.[12] also do not include information on achieving light exposure intervention goals in practice, omitting behaviour as a key ingredient for healthy light exposure.

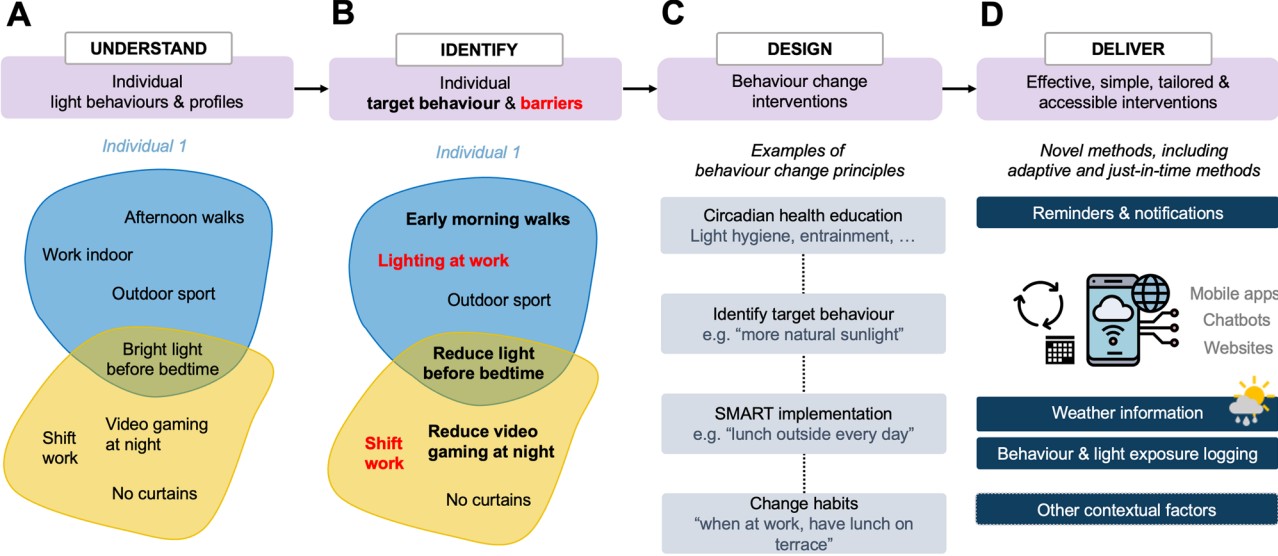

**Fig. 4 | A framework for identifying and delivering precision behavioural health.** The framework consists of four pillars including (**A**) understanding individual light behaviours and profiles (examples are for yellow and blue profiles of Fig. 2) to then (**B**) identify individual target behaviours and barriers that hinder optimal light exposure for circadian health. After these two individual steps, (**C**) individual behaviour change techniques embedded within tailored interventions need to be designed and (**D**) delivered in effective, simple, and accessible ways. These could also integrate external information sources such as whether data or wearable logging to give feedback on the fly to the user.

mood, visual comfort, circadian entrainment, or physical and mental health) at the same time without counteracting one endpoint by improving the other.

Finally, in terms of new methods and technologies, the scope of digital tools to implement behaviour change programmes and help habit formation must be evaluated. This also includes exploring the integration of feedback loops into the system, enabling the device to measure light behaviour, provide feedback on the user's behaviour, offer social support, and even deliver real-time recommendations based on factors such as upcoming weather or the user's daily schedule.

Especially this last idea calls for an interdisciplinary approach to develop such interventions further. Experts from health psychology and implementation sciences are needed to craft and develop effective interventions based on circadian neuroscience research and data sciences. Similar to our approach, recent efforts to deliver light interventions using an implementation science framework have identified barriers to implementation, which can now inform and enhance future interventions[105–108]. Computer scientists and engineers are needed to develop novel and user-friendly light loggers, implement, and integrate this into ecosystems and develop suitable apps or other means of delivery. The integrative

programme clearly highlights that interdisciplinary work is necessary to work towards achieving long-lasting behaviour changes that might be turned into healthy light exposure habits and delivered in an affordable way.

## Conclusion

Research on the non-image forming effects of light on human physiology and wellbeing has so far understood light largely as an object that can be crafted. Human-light interactions have thus implicitly been reduced to a passive exposure rather than an active behaviour. This article has proposed a novel framework that understands light exposure as such a behaviour – an interaction with the illuminated environment or light-emitting device that we actively engage with to meet our diverse needs. Viewing light exposure as a behaviour allows us to study light interaction in its complex, multi-faceted nature embedded within the location, culture and built environment of an individual. Research directions that arise from this framework include tools and devices to best log light exposure data, questionnaires that query human-light interactions to understand how people interact passively and actively with their daily environment, and new metrics to quantify such results. Individual behaviour change interventions to reach the target behaviour for circadian health of experiencing *bright days* and *dark nights* need to be

developed. While this may look like an extremely complicated endeavour, it is important to point out that often our theoretical possibilities of how we interact with light are limited by many external factors. Building on four pillars, as outlined in our implementation programme, we believe that core to achieving this is an individual approach, including key components of behaviour change techniques, and to delivering such interventions easily and accessibly through, for example, eHealth, mHealth and other digital methods.

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

## Acknowledgements
We thank Carolina Guidolin for sharing objective and subjective light exposure behaviour data and Johannes Zauner for visualisation of Fig. 2. We greatly thank the Daylight Academy (Velux Stiftung) for funding parts of this project. Priji Balakrishnan thanks the European Union's Horizon 2020 research and innovation programme for a Marie Skłodowska-Curie Individual Fellowship (grant agreement No. 101032279). Manuel Spitschan received financial support from the TUM Seed Fund and the Max Planck Society (Max Planck Free-Floating Research Group). The funders had no role in the preparation of the manuscript or decision to publish.

## Author contributions
Conceptualisation, A.M.B., P.B., M.S. Funding Acquisition, A.M.B., P.B., M.S. Methodology, A.M.B, P.B., M.S. Project administration, A.M.B. Supervision, M.S. Visualisation, A.M.B., P.B., M.S. Writing – Original draft, A.M.B. Writing – Review & Editing, A.M.B., P.B., M.S.

## Competing interests
The authors declare no competing interests.
