## [Peer Review File · Communications Psychology]

2nd Jul 24

Dear Anna,

I am sincerely sorry for the very protracted peer-review process of your submission "Behavioural determinants of physiologically-relevant light exposure". We have now obtained peer-review reports from two experts in the field, whose comments appear below. In light of their advice I am delighted to say that we are happy, in principle, to publish a suitably revised version in Communications Psychology under the open access CC BY license (Creative Commons Attribution v4.0 International License).

We will not send your revised paper for further review if, in the editors' judgement, the referees' comments on the present version have been addressed. If the revised paper is in Communications Psychology format, in accessible style and of appropriate length, we shall accept it for publication immediately. I have attached an edited version of your manuscript, and ask you to attend to each comment in detail.

EDITORIAL REQUESTS:

*Please attend to every item in the attached checklist to ensure that your manuscript complies at the point of resubmission.

* Please check whether your manuscript contains third-party images, such as figures from the literature, stock photos, clip art or commercial satellite and map data. If any of the display items in your manuscript (figures, tables, boxes or movies) include images that are the same as, or are adaptations of, previously published images, please fill in the Third Party Rights Table, and return to us when you submit your revised manuscript. This information will enable us to obtain the necessary rights to re-use such material. If we are unable to obtain the necessary rights to use or adapt any of the material that you wish to use, we will contact you to discuss alternative options.

* Communications Psychology uses a transparent peer review system. On author request, confidential information and data can be removed from the published reviewer reports and rebuttal letters prior to publication. If you are concerned about the release of confidential data, please let us know specifically what information you would like to have removed. Please note that we cannot incorporate redactions for any other reasons.

*If you have not done so already, please alert me to any related manuscripts from your group that are under consideration or in press at other journals, or are being written up for submission to other journals (see www.nature.com/authors/editorial_policies/duplicate.html for details).

FORMATTING GUIDELINES:

You will find a complete list of formatting requirements following this link:

<https://www.nature.com/documents/commsj-style-formatting-checklist-review-perspective.pdf>

Please use the checklist to prepare your manuscript for final submission. In the following, I also highlight some issues of particular importance.

** Figures

Please remove all figures from the main text and upload them individually, one figure per file. To ensure the swift processing of your paper please provide the highest quality, vector format, versions of your images (.ai, .eps, .psd) where available. Text and labelling should be in a separate layer to enable editing during the production process. If vector files are not available then please supply the figures in whichever format they were compiled in and not saved as flat .jpeg or .TIFF files. If your artwork contains any photographic images, please ensure these are at least 300 dpi.

* References

References appear as superscript Arabic numerals, in order of mention. The reference list mentions references in the numerical order in which they are mentioned in the main text. If a reference is cited more than once, the same number is used throughout the text and the reference receives a single entry in the reference list.

We ask that you select the most significant 5–10% of references in your list for highlighting, and add a single sentence in bold after each of these references to describe the main result and its significance.

Only papers that have been published or accepted by a named publication should be in the reference list (preprints and citations of datasets are also permitted). Unpublished/Submitted research should not be included in the reference list; it should only be mentioned briefly and

parenthetically in the main text. Note that no major arguments should rely on unpublished research.

Published conference abstracts and URLs for web sites should be cited parenthetically in the text, not in the reference list.

Footnotes are not used.

* Competing interests

Please include a "Competing interests" statement after the References. Note that we ask authors to declare both financial and non-financial competing interests. For more details, see <https://www.nature.com/authors/policies/competing.html>. If you have no financial or non-financial competing interests, please state so: "The authors declare no competing interests."

SUBMISSION INFORMATION:

* If you wish, you may also submit a visually arresting image, together with a concise legend, for consideration as a 'Hero Image' on our homepage. The file should be 1400x400 pixels and should be uploaded as 'Related Manuscript File'. In addition to our home page, we may also use this image (with credit) in other journal-specific promotional material.

* Your paper will be accompanied by a two-sentence editor's summary, of between 250-300 characters, when it is published on our homepage. Could you please approve the draft summary below or provide us with a suitably edited version.

Biller et al explain that humans actively shape their lighting environment through behaviour to meet specific individual needs. They propose that achieving healthy light exposure relies on shaping behaviour.

In order to accept your paper, we require the following:

- * A cover letter describing your response to our editorial requests.

- * A separate document detailing your point-by-point response to any issues raised by our referees (please include the referees' comments in this document).

- * The final version of your text as a Word or TeX/LaTeX file, with any tables prepared using the Table menu in Word or the table environment in TeX/LaTeX and using the 'track changes' feature in Word.

- * Production-quality versions of all figures, supplied as separate files. Photographic images should be 300 dpi in RGB format (.jpg, TIFF or native Photoshop format) and any labels/scale bars included in a separate layer from the image. Line art, graphs and schemes should be vector format (.ai, .eps, .pdf); Adobe Illustrator files are preferred and will minimize production time. Any chemical structures or schemes contained within figures should additionally be supplied as separate Chemdraw (.cdx) files.

At acceptance, the corresponding author will be required to complete an Open Access Licence to Publish on behalf of all authors, declare that all required third party permissions have been obtained.

Please note that your paper cannot be sent for typesetting to our production team until we have received this information; therefore, please ensure that you have this ready when submitting the final version of your manuscript.

ORCID

Communications Psychology is committed to improving transparency in authorship. As part of our efforts in this direction, we are now requesting that all authors identified as 'corresponding author' create and link their Open Researcher and Contributor Identifier (ORCID) with their account on the Manuscript Tracking System (MTS) prior to acceptance. ORCID helps the scientific community achieve unambiguous attribution of all scholarly contributions. For more information please visit <http://www.springernature.com/orcid>

For all corresponding authors listed on the manuscript, please follow the instructions in the link below to link your ORCID to your account on our MTS before submitting the final version of the manuscript. If you do not yet have an ORCID you will be able to create one in minutes.

IMPORTANT: All authors identified as ‘corresponding author’ on the manuscript must follow these instructions. Non-corresponding authors do not have to link their ORCID but are encouraged to do so. Please note that it will not be possible to add/modify ORCIDs at proof. Thus, if they wish to have their ORCID added to the paper they must also follow the above procedure prior to acceptance.

To support ORCID's aims, we only allow a single ORCID identifier to be attached to one account. If you have any issues attaching an ORCID identifier to your MTS account, please contact the Platform Support Helpdesk.

[link redacted]

We hope to hear from you within two weeks; please let us know if the process may take longer.

Best wishes,

Marike

Marike Schiffer, PhD

Chief Editor

Communications Psychology

REVIEWERS' EXPERTISE:

The reviewers are experts in circadian behaviour and physiology.

REVIEWERS' COMMENTS:

Reviewer #1 (Remarks to the Author):

The manuscript by Biller et al. reviews the science behind how lighting affects human physiology and proposes a framework for implementing lighting interventions in real-world settings. The topic is important and raises important considerations for researchers and practitioners to consider before deploying lighting interventions. However, there are two issues that the authors should address:

First, please address the importance of the pattern of light exposure (line 131-132). Several researchers have conducted studies showing that intermittent light pulses and flashes can influence circadian phase shifting. Dr. Spitschan is an author on a few of these papers but there are others too. Pattern seems to be distinctly separate from duration, intensity, and timing because of the regeneration of photoreceptor cells that occurs between light pulses. This should be included and discussed because it represents a potential mode of intervention.

Second, the authors mention implementation science at the end of the paper but failed to mention any of the work that has already been done to deliver light interventions using an implementation science framework (e.g., Harrison et al. 2020, Bessman et al. 2023). These studies should be discussed. The authors should also acknowledge parallel efforts to identify the barriers to implementation of lighting interventions (e.g., Faulkner et al. 2019, 2020). This is not a comprehensive list and there may be other relevant work.

I also identified a number of typos:

Page 2, line 31 and page 3, line 70, I think there is a typo. Do you mean culminated instead of cumulated?

Page 3, line 73, I think this is an incomplete sentence. Should this say, "...humans should experience bright days..." or something similar?

Page 5, line 125, I think you need the article "the" before "right light."

Page 6, line 130, please add the word "sleep" before "onset" for clarity.

Line 197, "unhandy" is an unusual word to use here. I'm not sure I fully understand what it means in this context. Do you mean "unhelpful?"

204, I think you mean "are" not "or"

246, I think you mean "averting" instead of "reverting"

Reviewer #2 (Remarks to the Author):

I was pleased to review this submission, which is well argued and an important and widely relevant topic.

My main comments are minor, and particularly around some English usage points.

1. Line 31 and line 70: culminated, rather than cumulated

2. Figure 1C: the figure shows the pathway from the SCN to pineal, via superior cervical ganglion. However, this is not explained in legend. If the authors consider this pathway relevant to their argument, I would suggest adding a brief explanation in the legend. If not relevant, I would suggest removing it from the figure.

3. Line 125: 'right light at the right time' - suggest elaborating a little more - e.g. right quality and quantity of light, at the right time?

Lines 141 - 147: it would be helpful to add a little more detail on what we know of the anatomy of NIF pathways from the retina to other areas involved in arousal and sleep/wake regulation, in addition to the retinal hypothalamic tract (e.g. retina to VLPO and other arousal nuclei).

4. Line 148: Metrology - suggest change to measurement, as metrology is a rather obscure term.

5. Line 209 and line 210: phase delays IN circadian rhythms and delays IN sleep

6. Section 'A framework for light exposure behaviours' (lines 267 - 321):

- the authors mention location/ geography with respect to latitude, climate and the built environment. Another important factor is time zones - e.g. sleep wake and light exposure behaviours in Spain are influenced by the fact that the country is in the western end of the European time zone.. If the authors agree, would suggest adding mention of this.

- Further to argument that light exposure is an active, and not just passive, dimension, brief mention could also be made of how certain disorders interact potently with light exposure behaviours. E.g. many individuals with mental disorders have reduced levels of motivation, reduced physical activity, sleep disturbances, which reduce light exposure behaviours during daytime, and increases night-time light. In turn, this contributes to further sleep-circadian dysregulation, worsening symptoms and function further.

Reviewer #1 (Remarks to the Author):

The manuscript by Biller et al. reviews the science behind how lighting affects human physiology and proposes a framework for implementing lighting interventions in real-world settings. The topic is important and raises important considerations for researchers and practitioners to consider before deploying lighting interventions. However, there are two issues that the authors should address:

First, please address the importance of the pattern of light exposure (line 131-132). Several researchers have conducted studies showing that intermittent light pulses and flashes can influence circadian phase shifting. Dr. Spitschan is an author on a few of these papers but there are others too. Pattern seems to be distinctly separate from duration, intensity, and timing because of the regeneration of photoreceptor cells that occurs between light pulses. This should be included and discussed because it represents a potential mode of intervention.

We agree with the reviewers point and have amended the paragraph as follows (references only included in article):

Since light suppresses the release of melatonin, the timing (when), duration (how long), **temporal pattern (how light is structured over time, e.g., flickering)**, intensity (how much illuminance), and spectrum (which wavelengths and spectral composition) of light exposure can influence **photoentrainment**.

Later on, we now include temporal pattern (references only included in article):

Since the combination of these four light exposure parameters (timing, duration, **temporal pattern, intensity**, and spectrum) play a major role in shaping our physiological output, quantification of these attributes of light is necessary to measure their NIF effect, **as in real-world scenarios, light exposure patterns are highly complex**.

Second, the authors mention implementation science at the end of the paper but failed to mention any of the work that has already been done to deliver light interventions using an implementation science framework (e.g., Harrison et al. 2020, Bessman et al. 2023). These studies should be discussed. The authors should also acknowledge parallel efforts to identify the barriers to implementation of lighting interventions (e.g., Faulkner et al. 2019, 2020). This is not a comprehensive list and there may be other relevant work.

We thank the reviewer for making us aware of these efforts. We have incorporated them as follows in lines 477-479:

Similar to our approach, recent efforts to deliver light interventions using an implementation science framework have identified barriers to implementation, which can now inform and enhance future interventions.

I also identified a number of typos:

Page 2, line 31 and page 3, line 70, I think there is a typo. Do you mean culminated instead of cumulated?

Page 3, line 73, I think this is an incomplete sentence. Should this say, "...humans should experience bright days..." or something similar?

Page 5, line 125, I think you need the article "the" before "right light."

Page 6, line 130, please add the word "sleep" before "onset" for clarity.

Line 197, "unhandy" is an unusual word to use here. I'm not sure I fully understand what it means in this context. Do you mean "unhelpful?"

204, I think you mean "are" not "or"

246, I think you mean "averting" instead of "reverting"

We thank the reviewer very much for carefully reading through the manuscript and identifying the typos which are now resolved.

Reviewer #2 (Remarks to the Author):

I was pleased to review this submission, which is well argued and an important and widely relevant topic. My main comments are minor, and particularly around some English usage points.

1. Line 31 and line 70: culminated, rather than cumulated

2. Figure 1C: the figure shows the pathway from the SCN to pineal, via superior cervical ganglion. However, this is not explained in legend. If the authors consider this pathway relevant to their argument, I would suggest adding a brief explanation in the legend. If not relevant, I would suggest removing it from the figure.

We thank the reviewer for their thoughtful comments. We have resolved all typos accordingly and expanded the figure legend of panel C of Figure 1 which now reads as follows:

C Retinohypothalamic non-image-forming pathway. The melanopsin/ipRGC-mediated pathway connects the retina to the hypothalamus, and more specifically, the suprachiasmatic nucleus (SCN). The pathway from the SCN to the pineal gland involves a multi-synaptic route: the signal travels from the SCN to the intermediolateral cell column of the spinal cord, then to the superior cervical ganglion (SCG), which is part of the sympathetic nervous system. From the SCG, the signal reaches the pineal gland (PG) to trigger melatonin release in response to darkness.

3. Line 125: 'right light at the right time' - suggest elaborating a little more - e.g. right quality and quantity of light, at the right time?

We have amended the sentence as follows:

A main consequence of human exposure to the right light (**quality, quantity**) at the right time (**of day, across the year**) is circadian photoentrainment which describes the synchronisation of internal (circadian) time and external (clock) time.

Lines 141 - 147: it would be helpful to add a little more detail on what we know of the anatomy of NIF pathways from the retina to other areas involved in arousal and sleep/wake regulation, in addition to the retinal hypothalamic tract (e.g. retina to VLPO and other arousal nuclei).

We amended the following text passage (references only included in article):

Light also directly impacts sleepiness and alertness levels through ascending arousal systems **in the hypothalamus** reaching the cortex (**e.g., tuberomammillary nucleus, locus coeruleus, raphe nuclei**): bright high-melanopic light acutely increases alertness levels while dim low-melanopic light contributes to sleepiness thereby serving as an important **modulator** for cognitive performance. **This arousal system is influenced by inhibitory actions of the ventrolateral preoptic nucleus (VLMP): during wakefulness,**

arousal systems inhibit the VLPO, while during sleep, the VLPO inhibits these arousal systems to maintain sleep states. There is also emerging evidence that light exposure prior to bedtime or even during the day can also *directly* affect sleep quality as well as sleep architecture (i.e., amount of specific sleep stages). Other NIF effects of light include pupil constrictions (through the ipRGCs' trajectory to the olivary pretectal nucleus [OPN] which sends signals to the autonomic nervous system), heart rate and core body temperature modulation (through the autonomous nervous system), and cortisol production (through the SCN's effect on the hypothalamic-pituitary-adrenal axis).

4. Line 148: Metrology - suggest change to measurement, as metrology is a rather obscure term.

We changed the title of the paragraph:

How can we measure the ipRGC-influenced responses to light?

We also amended the sentence in lines 445-448:

In addition, we need to keep improving measurements (metrology) and developing metrics further that quantify NIF effects of light and determine which metrics that describe light exposure are relevant and useful for the individual.

5. Line 209 and line 210: phase delays IN circadian rhythms and delays IN sleep

This is resolved.

6. Section 'A framework for light exposure behaviours' (lines 267 - 321):

- the authors mention location/ geography with respect to latitude, climate and the built environment. Another important factor is time zones - e.g. sleep wake and light exposure behaviours in Spain are influenced by the fact that the country is in the western end of the European time zone.. If the authors agree, would suggest adding mention of this.

We agree with this notion and amended the section as follows:

While we can also interact and shape the built environment and, at least to some extent also culture, we have little to no impact on external factors such as the location within a time zone (including clock time changes), photoperiod, sunshine hours, local climate, or ambient temperature (Figure 3A). These factors have historically largely influenced and shaped our culture (e.g., customs, festivities, or norms) and have dictated the built environment (e.g., building form, window size and openings, proximity to other buildings, street orientations, materiality, or lighting significance), including its reciprocal interaction. Shaped by these determinants are our own behaviours which

include how we perceive light (valence), how we interact with light (affordances), what lifestyle we choose (hobbies, jobs, **pets**) and **what temporal pattern this lifestyle has**, or what individual preferences we display towards light (e.g., sunseeker vs sun avoiders).

- *Further to argument that light exposure is an active, and not just passive, dimension, brief mention could also be made of how certain disorders interact potently with light exposure behaviours. E.g. many individuals with mental disorders have reduced levels of motivation, reduced physical activity, sleep disturbances, which reduce light exposure behaviours during daytime, and increases night-time light. In turn, this contributes to further sleep-circadian dysregulation, worsening symptoms and function further.*

This is an important idea and we incorporated it as follows in lines 308-313:

Furthermore, certain disorders or health conditions interact can with individual behaviour and can have a large impact on individual light exposure profiles potentially leading to decreased (e.g., bright light sensitivity, xeroderma pigmentosum, anxiety, depression, severe obesity, gaming addiction, etc.) or increased (e.g., Vitamin D deficiency or seasonal affective disorder) sunlight exposure.